# Thermal Annealing Effect on Surface-Enhanced Raman Scattering of Gold Films Deposited on Liquid Substrates

**DOI:** 10.3390/molecules28031472

**Published:** 2023-02-03

**Authors:** Ziran Ye, Haixia Huang, Fengyun Xu, Ping Lu, Yiben Chen, Jiawei Shen, Gaoxiang Ye, Fan Gao, Bo Yan

**Affiliations:** 1Department of Applied Physics, College of Science, Zhejiang University of Technology, Hangzhou 310058, China; 2Center for Optics & Optoelectronics Research (COOR), Collaborative Innovation Center for Information Technology in Biological and Medical Physics, College of Science, Zhejiang University of Technology, Hangzhou 310058, China; 3Department of Physics, Zhejiang University, Hangzhou 310015, China

**Keywords:** thin films, annealing, SERS, nanoparticles, liquid substrates

## Abstract

We prepare metal films with various thicknesses on liquid substrates by thermal evaporation and investigate the annealing effect on these films. Gold films deposited on a silicone oil surface consist of a large number of branched aggregates, which contains plenty of gold nanoparticles. This characteristic morphology is mainly attributed to the isotropic and free-sustained liquid substrate. Thermal annealing results in the reintegration of nanoparticles; thus, the surface morphology and microstructure of gold films change significantly. The dependence of annealing conditions on the surface-enhanced Raman scattering performance of gold films is studied, in which gold films show favorable Raman activity when annealed at certain annealing temperature and the experimental results are verified by simulation analysis. The study on the optimal annealing temperature of surface-enhanced Raman scattering substrate will pave the way for the potential application of films deposited on liquid surfaces in microfluidics and enhanced Raman detection.

## 1. Introduction

Thin film deposition on liquid substrates has attracted tremendous attention in the past two decades due to the unique growth mechanism and surface morphology compared with films grown on solid substrates [1,2,3,4]. Ye et al. deposited silver films on the surface of silicone oil by thermal evaporation, through which a simple and feasible method for preparing metal films on liquid substrates was proposed [5]. After that, various metal materials such as gold (Au), silver (Ag), copper (Cu), zinc (Zn), iron (Fe) and aluminum (Al) were deposited on different liquid substrates (silicone oil, deionized liquid, etc.) [5,6,7,8,9,10]. These studies found that the ramified structures in metal films deposited on liquid surfaces consist of substantial nanoparticles, and the diameter of particles is related to the deposition conditions, which is also closely related to the properties of metals. For instance, the particle size of Al metal changes little with the nominal film thickness and deposition rate [11], whereas the mean diameter of nanoparticles in Ag film decreases with the increase of deposition thickness, apparently [5], etc. By varying the deposition parameters during the growth process, the characteristic morphology and growth mechanism of thin films deposited on liquid substrates have been systematically studied. The growth mechanism basically conforms to the two-stage growth model: the metal atoms first nucleated on the liquid surface and then formed quasi-circular aggregates; in the second stage, the aggregates diffused freely on the liquid surface due to random movement and formed ramified structures after collision and rotation [5,12]. This characteristic growth can be mainly attributed to the free-sustained and isotropic liquid substrates, on which the tangential force between the liquid substrates and the metal films is less; thus, the lattice mismatch can hardly happen. However, because of the surface effect of the liquid phase substrates, a large number of voids and defects are produced inside the metal clusters. Zhang et al. used transmission electron microscopy (TEM) to study the crystallization properties of silver films deposited on the surface of silicone oil at different substrate temperatures; it can be found that the atomic arrangement in silver island became highly ordered and the crystallinity was increased with the continuous increase of substrate temperature, which effectively eliminated some of the defects in the island [13].

Since surface-enhanced Raman scattering (SERS) was observed in the 1970s, it has received many impressive scientific achievements in related applications of biology, chemistry and physics [14,15,16]. The chemical bond vibration of probe molecules near the tip or gap of metal nanoparticles is detected by SERS spectrum, which can be used as a tool for effective biomolecular analysis, and the detection limit can reach the level of a single molecule at present [17]. It is generally accepted that the enhancement of this local field is caused by the excited surface–plasmon resonance [18]. The enhancement ability of SERS substrates depends on the “hot spots” formed among metal nanoparticles, which can provide a strong local field for Raman detection [19,20]. The formation of “hot spots” is highly dependent on the size, spacing, shape, density, and surrounding medium environment of nanoparticles [19,21]. Over the past few decades, the fabrication technique of SERS substrate has been extensively studied by various methods, such as the template process [22], the free corrosion method of alloy [23], the nano-lithography process [24], methods for the chemical synthesis of nanomaterials with special structures [25], etc. Furthermore, the optimization of SERS performance has also attracted much attention, among which annealing was found to be an effective method in the past research [19,22,23]. By adjusting the thermal annealing conditions and methods, the structure and size of nanoparticles can be controlled with reduced levels of impurities.

The annealing process is a commonly used technical means of heat treatment. It can be used to eliminate residual internal stress and lattice defects, reduce film surface energy, bond between film layers, etc., and thus has an important impact on the physical and chemical characteristics (such as microscopic morphology, crystallinity, mechanical properties, electrical properties and optical properties, etc.) of the deposited films [26,27,28,29,30,31,32]. Zhou et al. annealed the Au film based on electron beam deposition at 100 °C to 400 °C and analyzed the evolution of residual stress in the Au film after annealing; they found that the voids in the Au film were eliminated, the stability of the film increased and the stress level was also raised [33]. Zhang et al. heated polystyrene core/Au nanoparticle shell (PS@Au) and found that the Au particle size and morphology varied after 200 °C and 400 °C annealing, which had favorable application prospects in R6G and crystal violet detection [19]. In general, the annealing process includes three stages: slow heating up, maintenance at a constant temperature, and cooling action. Thermal annealing is conducive to the diffusion and nucleation of separated crystals on the substrate surface to grow into uneven crystals at high temperature, as well as improving the structural defects and varying the grain size, pore size and grain boundaries [34,35,36]. 

In this work, we prepared Au films on silicone oil surfaces by thermal evaporation, and then the prepared samples were annealed to improve the film morphologies and defects. The microstructure and growth mechanisms of Au films at various annealing conditions were systematically investigated. In addition, we utilized this method of depositing Au films on liquid surfaces to prepare SERS substrates, and the SERS performance of Au films annealed at different temperatures was evaluated.

## 2. Results and Discussion

Optical images in Figure 1 exhibit the morphologies of Au films deposited on silicone oil surfaces with nominal thicknesses of (a) 5 nm, (b) 10 nm and (c) 20 nm, in which a large number of branched Au aggregates can be clearly observed. The viscosity of silicone oil is about 175 cSt at 25 °C, and the silicone oil can be regarded as a free-sustained substrate with isotropic characteristics, on which the growth of Au atoms obeys the two-stage growth model [5]: (1) Au atoms deposited on the surface of silicone oil nucleate into circular clusters; (2) the circular clusters form ramified aggregates through Brownian motion and collision. As shown in Figure 1, the mean diameter of Au aggregates and film coverage increases significantly with film thickness, whereas the gap width decreases gradually. Finally, the ramified aggregates were connected to each other to form a continuous Au film. If the substrate temperature increases, the surface tension of silicone oil and the viscosity decrease, which will weaken the interaction between Au film and silicone oil, and the Au atom tends to form a three-dimensional (3D) island rather than a branching structure [13]. We transferred the Au film on the silicon oil surface to the solid substrate before annealing, and the morphological change of Au films can be further observed after annealing. Optical images in Figure 2 exhibit the morphologies of 10 nm Au films without annealing (Figure 2a), and annealed at 150 °C, 250 °C, and 350 °C (Figure 2b–d) each for 1 h, respectively. As can be seen from Figure 2, Au films annealed at various temperatures exhibited ramified structures, with the gap width increasing slightly as the annealing temperature increased.

We further investigated the influence of thermal annealing on the microstructure of Au films. SEM images of Au films with different thicknesses were shown in Figure 3, in which (a), (b) and (c) exhibit the morphology of Au films deposited on silicone oil surfaces with film thickness of 5 nm, 10 nm and 20 nm, respectively. The images in Figure 3a–c further reveal the film morphology under various annealing conditions of (i) without annealing and annealed at (ii) 150 °C, (iii) 250 °C and (iv) 350 °C for 1 h. Typical morphological evolution can be seen from Figure 3a, in which Au films consist of substantial Au nanoparticles, the assemblage of Au nanoparticles in 5 nm Au film was gradually disconnected as the annealing temperature increased, while most of the Au particles clustered tended to form large aggregates. Due to the size effect, the melting point of Au nanoparticles is lower than that of solid Au [37], and the melting point will change with the size of the nanoparticles. During the annealing treatment, small nanoparticles aggregated into larger nanoparticles and the melting point increased accordingly. Therefore, the size of Au nanoparticles increased with the annealing temperature. Figure 3b exhibits that Au nanoparticles formed adherent clusters in 10 nm Au film when the annealing temperature was 150 °C. However, when annealed at 250 °C, the previously closely packed clusters became gradually separated and exhibited an irregular striped structure, with more tiny Au particles forming at the edge of cluster assembly. When the annealing temperature further increased to 350 °C, larger Au particles were produced with uneven size distribution. For the 20 nm Au films illustrated in Figure 3c, there was a tightly bonded mesh structure when annealed at 150 °C. In Figure 3(ciii), the gap width between the tightly bonded structures increased and small nanoparticles with irregular shape appeared at the edges when the annealing temperature was 250 °C. The large aggregates further separated, disconnected, and ruptured as the annealing temperature increased to 350 °C. Moreover, in order to clearly depict the magnitude variation of Au nanoparticles under different annealing conditions, the diameters of randomly selected 50 Au particles in 5 nm Au film annealed at different temperatures were measured, as exhibited in Figure 3d. The diameter of Au nanoparticles was mainly between 40 nm and 90 nm for various annealing temperatures, and the mean diameter of Au nanoparticles increased significantly from 45.1 nm to 75.2 nm with the annealing temperature (Figure 3(div)), which may be due to the coalescence of small nanoparticles during the annealing process.

As the annealing temperature further increased to 600 °C, the morphologies of Au films with various thicknesses observed by SEM were shown in Figure 4, in which Figure 4a–c correspond to Au films with thicknesses of 5 nm, 10 nm and 20 nm, respectively. It can be indicated that the Au films annealed at 600 °C mainly consist of substantial spherical nanoparticles, which may be attributed to the enhanced spheroidization at higher annealing temperatures. The annealing process can modify the crystallinity of the as-prepared films and further improve the crystallinity at high annealing temperatures.

It can be indicated from the results that Au films distributed widely and densely on the surface of the substrate before annealing. In the annealing process, the grains melted into a thin liquid layer first, and the effect of surface tension caused the tendency of automatic shrinkage of the liquid surface, which resulted in the different degrees of spheroidization of the Au films. When the annealing temperature increased, the transition from thin liquid layer to sphere happened as a spontaneous process. Therefore, the microstructure of Au films strongly depended on the annealing temperature. When the annealing temperature was low, the migration of Au atoms and the shrinkage of the bulges on the surface of the Au film were relatively slow, resulting in poor spheroidization. Therefore, structures of irregular clusters, strips and even networks appeared. It is easier to form spherical Au nanostructures when annealed at a higher temperature. Under the same annealing conditions, the bulges formed on the surface of the thinner Au film were smaller; thus, spherical structures were more likely to be formed in thinner films. The evolution of the morphology of Au films under various annealing conditions can be depicted as follows: Au films began to deform at a low annealing temperature. As the annealing temperature increased, the ramified aggregates in Au films gradually deformed into discrete irregular particles and finally spherical particles. The spheroidization of the films mainly aimed for the minimization of surface free energy.

Comparing with metal films grown on solid substrates, the irregular arrangement of aggregates in films deposited on liquid substrates can hardly exhibit crystallinic properties. In general, high-temperature annealing treatment results in the atomic rearrangement of Au aggregates, which realizes void elimination and leads to improvement in crystallinity [33]. In this work, the impurities on the surface and the defects inside the Au films can be significantly decreased as the annealing temperature increased, which led to better crystallinity.

SERS measurement was conducted to explore the influence of thermal annealing on the enhancement of Raman signals on the as-prepared Au films. Au films with nominal thicknesses of 5 nm, 10 nm and 20 nm annealed at 150 °C, 250 °C and 350 °C were selected as SERS substrates in the experiment, and R6G was used as a probe molecule for SERS detection. SERS spectra of all the samples are exhibited in Figure 5a–c, in which the characteristic peaks of R6G can be clearly distinguished at different annealing temperatures. These characteristic peaks in Raman spectra correspond to different vibration modes of R6G molecule [38]: the peaks at 1360 cm^−1^, 1508 cm^−1^, and 1647 cm^−1^ referred to the C-C stretching vibration modes of the aromatic ring; the peaks at 1120 cm^−1^ and 1180 cm^−1^ correspond to the in-plane bending vibration modes of C-H; the bands at 1308 cm^−1^ and 1573 cm^−1^ are related to the N-H in-plane bend modes; and the peak at 771 cm^−1^ is a result of the C-H out-of-plane bending vibration mode, while the peak at 611 cm^−1^ is assigned to the C-C-C in-plane bending vibration mode. In addition, considering that the silicone oil may has not be thoroughly removed, thus the peaks at 1120 cm^−1^ and 1180 cm^−1^ in silicone oil may affect Raman detection, we mainly compared the Raman intensity at 1508 cm^−1^. Figure 5d indicates the relationship between the characteristic peak intensity of 1508 cm^−1^ and the annealing temperature, in which the Au films annealed at 150 °C and 250 °C showed more enhanced SERS performance compared with Au films without annealing. In general, localized surface plasmon resonance (LSPR) is one of the most important optical properties of precious metal nanoparticles. The enhancement of Raman signals mainly originates from the LSPR effect of metal nanostructures, which is highly dependent on the size, shape and spacing of nanoparticles. According to the morphology of the Au films in the SEM images under different annealing conditions, more spherical Au nanoparticles were obtained when annealed at 150 °C and 250 °C, and parts of small Au particles were merged into larger particles, which may be one of the reasons for the enhancement of Raman intensity. Au nanoparticles with a size of ~80 nm have a resonance peak at the excitation wavelength of ~633 nm, with the intensity of the plasmon resonance significantly enhanced [39]. Therefore, when the size of Au nanoparticles increased, the resonance peak red-shifted towards the excitation wavelength, resulting in the enhanced Raman intensity. However, the SERS intensity decreased when the annealing temperature further increased to 350 °C. We speculate that the gaps between Au nanoparticles exceed the distance of electromagnetic coupling, and the resonance peak was blue-shifted, which will cause the absence of partial nanoparticles couplings. According to Figure 5d, the enhancement factor (*EF*) can be calculated according to the definition formula [40,41]:(1)$$\;EF=\left(I_{SERS}/N_{SERS}\right)/\left(I_{0}/N_{0}\right)$$
where $I_{SERS}$ is the Raman signal intensity of probe molecules attached to the substrate in a specific vibration mode, here we chose 1508 cm^−1^ Raman peaks to calculate *EF*. $I_{0}$ is the normal Raman signal intensity of the R6G; $N_{SERS}$ and $N_{0}$ are the number of R6G molecules on the SERS substrate and a silicon wafer in the laser spot, respectively. Moreover, the number N of R6G for the influence signal was estimated by the following formulas [12,42]:(2)$$N=\frac{N_{A}A_{laser}CV}{A_{bulk}}$$
where C represents the molar concentration of R6G and V represents the volume of R6G solution; $N_{A}$ is Avogadro constant; $A_{bulk}$ indicates the area of R6G distribution; and $A_{laser}$ is the area of laser spot. The preparation method of R6G samples adsorbed on the substrate and measured in the laser spot was identical with controlled measurement conditions. Therefore, $N_{SERS}/N_{0}\approx C_{SERS}/C_{0}$ (where $C_{0}$ is the concentrations of R6G on the silicon wafer and $C_{SERS}$ indicates the concentrations of R6G on the substrates) [42]. According to the above formulas, we found that the SERS enhancement factor of branched Au aggregates prepared on liquid substrate in this experiment can reach the order of ~10^6^.

In order to study the correlation between the annealing temperature of Au films and the electromagnetic field, the FDTD method was used to simulate the spatial distribution of electromagnetic field intensity. SEM images of Au films with nominal film thicknesses of 10 nm were used to model the structure of Au films under various annealing conditions. The simulation parameters in the FDTD simulation were set as follows: the simulation time was set to 1 ps and the calculation accuracy was adjusted to 3; the boundary condition was set as the perfectly matched layer (PML); the excitation light wavelength was 633 nm; and the monitoring light source was set as the total field scattered field (TFSF). Figure 6a–d reveal the electromagnetic field distributions of Au films with nominal film thickness of 10 nm without annealing and annealed at 150 °C, 250 °C and 350 °C, respectively. The electric field enhancement amplitude was set to be coincident to show the electric field distribution of Au films under various annealing conditions. The effect of different annealing temperatures on SERS activity was systematically studied, and the simulation results showed that there were a large number of “hot spots” distributed on the surface of Au films deposited on silicone oil surfaces, which were extremely sensitive to Raman signals, resulting in the excellent SERS performance. Furthermore, the number of hot spots gradually increased with the annealing temperature until reaching 250 °C, as shown in Figure 6a–c. The hot spots of the sample without annealing treatment were dispersed, and became larger and more densely distributed with the increase of annealing temperature. It is speculated that, at lower annealing temperature, the electric field enhancement of Au nanoparticles was mainly dependent on the size variation of the nanoparticles. When the annealing temperature reached 350 °C, the amount of “hot spots” in Au films decreased in the simulation results (Figure 6d), which was consistent with the measured Raman spectrum. We used the EM field distribution to calculate the enhancement factor *EF*, which is usually defined as *(E/E*_0_*)*^4^, where *E* is the excitation electric field and *E*_0_ is the input electric field, and *EF* is proportional to *E/E*_0_ [43]. According to the simulation results, the electric field of the hot spots in the red region was significantly enhanced, where the simulation results show a value of (*E/E*_0_)_max_ > 10; thus the enhancement factor can be ~10^4^ in this model. The SERS substrate prepared in this experiment had a favorable performance, and we will continue to discuss the influence of the surface tension of silicone oil on the morphology of Au film in future work and further optimize the substrate performance by adjusting annealing conditions and film thickness.

## 3. Materials and Methods

In this experiment, Au films deposited on liquid surfaces were prepared by thermal evaporation at room temperature (T = 20 ± 3 °C), as illustrated in Figure 7. Frosted glass wafers, polished glass wafers and silicon wafers (10 × 10 mm^2^) were cleaned ultrasonically with acetone, ethanol and deionized water, each for 5 min, respectively. A droplet of silicone oil (DOW CORNING 705 Diffusion Pump Fluid) was then dropped onto the surface of the frosted glass wafer; the silicone oil was coated evenly on the surface of the frosted glass wafer with the thickness of about 0.5 mm. A Au wire (99.999%, Sinopharm Chemical Reagent Co., Ltd., Shanghai, China) was wounded on a U-shaped tungsten wire in the vacuum chamber as the source for thermal evaporation, and the samples were placed about 25 cm under the tungsten wire. A molecular pump was utilized to vacuumize the chamber until the pressure in the chamber was lower than 3 × 10^−4^ Pa. Afterwards, we deposited Au films of various thicknesses onto the surface of silicone oil under the deposition rate of 0.1 Å/s by adjusting the driving current. The deposition rate and nominal thickness of Au films were in situ monitored and recorded by a quartz-crystal balance (CRTM 8000) placed near the samples. After deposition, the films were kept in the vacuum chamber for 15 min stabilization.

The as-prepared Au films were then taken out from the vacuum chamber for film transfer. Each sample was first placed into a glass Petri dish and covered with a polished glass wafer for 10 min, and then soaked in acetone for 10 min for film transfer. Next, the polished glass wafer was separated from the frosted glass wafer carefully with a tweezer, after which the Au films were transferred to the surface of the polished glass wafer. Finally, the samples were soaked in acetone and ethanol for 5 min each, respectively, to remove the residual silicone oil, and were then dried at 50 °C on a hotplate. After the film transfer, the as-prepared Au films were annealed in a quartz tube furnace (OTF-1200X, Hefei Kejing Materials Technology Co., Ltd., Hefei, China). In our experiment, the Au films were annealed at 150 °C, 250 °C, 350 °C and 600 °C for 1 h, respectively. In addition, we adopted the heating rate of 10 °C/min to avoid the overshoot of temperature due to the fast heating rate. 

For the characterization of the surface morphology of Au films deposited on liquid surfaces, an optical microscope (Leica DMLM) was used for morphological observation and then high-resolution scanning electron microscopy (SEM, Zeiss-SuprATM55) was utilized to obtain detailed microstructures in Au films.

For the investigation of the SERS performance of Au films deposited on liquid surfaces, Au films with nominal thicknesses of 5 nm, 10 nm, 20 nm were selected for Raman measurement. Annealing temperatures of 150 °C, 250 °C and 350 °C were chosen to study the dependence of annealing effect on SERS performance. Rhodamine 6G(R6G) was used as the probe molecule for SERS detection; Au films were soaked in R6G solution with a concentration of 10^−7^ M for 12 h and then dried naturally. SERS spectra of all the samples were observed by a confocal Raman microscope (Renishaw inVia) with the excitation wavelength of 633 nm and the laser power of 0.85 mW. Moreover, it was necessary to preheat the laser for 15 min to stabilize the laser power before the measurement. During the measurement, a 50 × L telephoto lens was used to collect the Raman spectrum, the spot diameter of the objective lens was 1.5 μm, the exposure time was 10 s and the cumulative number was 2.

## 4. Conclusions

In summary, we prepared Au films with different thicknesses on liquid surfaces as SERS substrates by thermal evaporation, and studied the effect of thermal annealing treatment. The width of the branched aggregates in Au films increased and the gap width decreased to form the compact films with the increased nominal film thickness. The effects of thermal annealing on the morphology and SERS performance of Au films were studied systematically. The results showed that the SERS sensitivity of Au films showed dependence on the annealing temperature; Au films annealed at 150 °C and 250 °C exhibited excellent SERS performance. The electromagnetic field distributions in Au films at various annealing temperatures were simulated by the FDTD method, and abundant “hot spots” were distributed in the prepared Au films, which validated the experimental results. This study shows great potential of metal films deposited on liquid surfaces as promising candidates in the application of SERS detection. The optimization of annealing condition will facilitate the development of these films as optical-based biosensors in sensitive chemical detection and biomedical analysis.

## Figures and Tables

**Figure 1 molecules-28-01472-f001:**
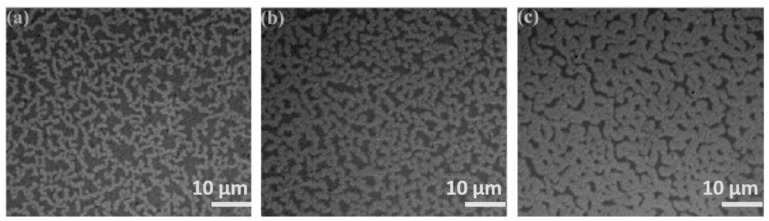
Morphology of Au films deposited on silicone oil surfaces with film thickness of (**a**) 5 nm, (**b**) 10 nm, and (**c**) 20 nm, respectively.

**Figure 2 molecules-28-01472-f002:**
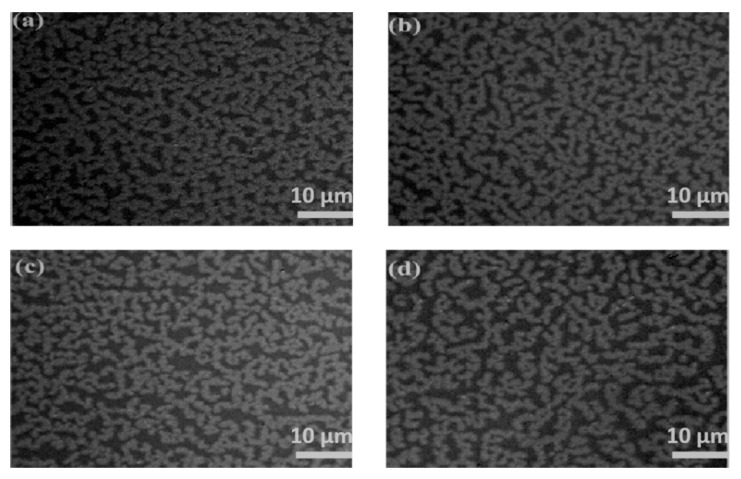
Morphology of 10 nm Au films deposited on silicone oil surfaces (**a**) without annealing and annealed at (**b**) 150 °C, (**c**) 250 °C, (**d**) 350 °C, respectively.

**Figure 3 molecules-28-01472-f003:**
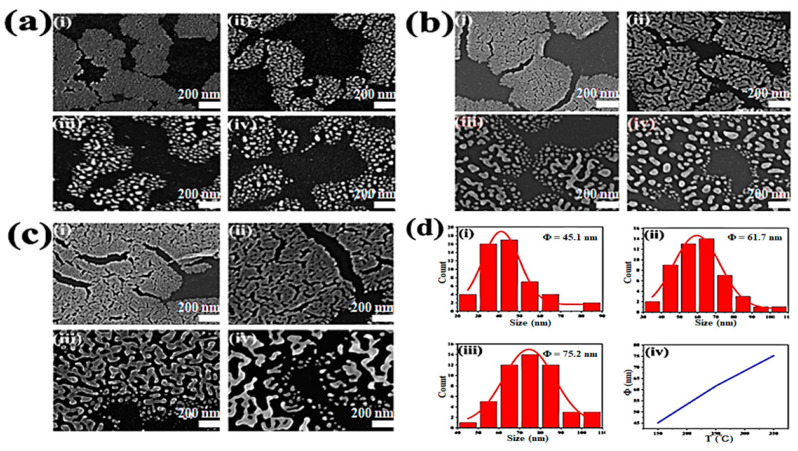
SEM images of Au films deposited on silicone oil surfaces with film thickness of (**a**) 5 nm, (**b**) 10 nm, and (**c**) 20 nm, in which (**i**–**iv**) represent Au films without annealing and annealed at 150 °C, 250 °C and 350 °C for 1 h, respectively. (**d**) Diameter distribution of nanoparticles in 5 nm Au films at (**i**) 150 °C, (**ii**) 250 °C, and (**iii**) 350 °C, and (**iv**) exhibits the relationship between annealing temperature and mean diameter of Au nanoparticles.

**Figure 4 molecules-28-01472-f004:**
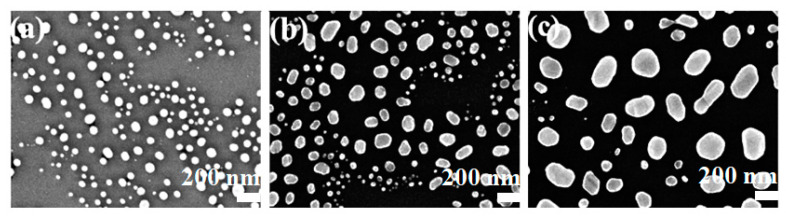
SEM images of Au films deposited on silicone oil surfaces with film thicknesses of (**a**) 5 nm, (**b**) 10 nm, and (**c**) 20 nm annealed at 600 °C for 1 h.

**Figure 5 molecules-28-01472-f005:**
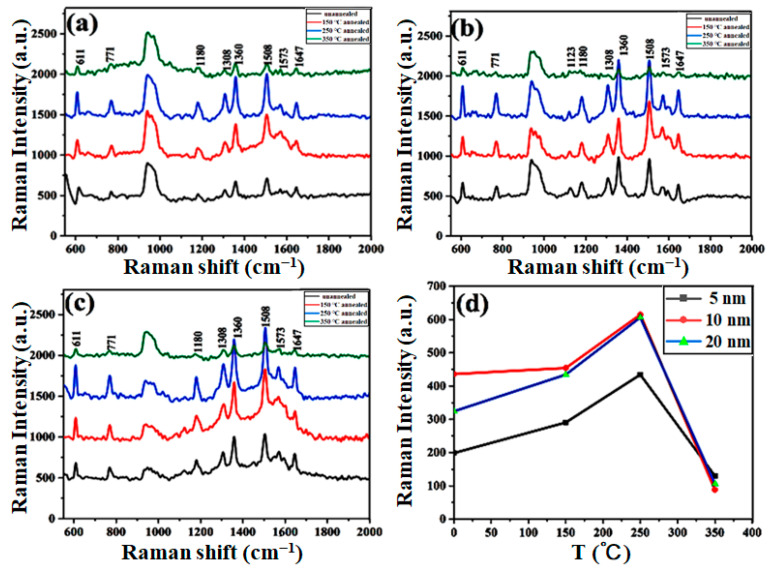
Raman spectra of 10^−7^ M R6G on Au films with various film thicknesses under different annealing temperatures: (**a**) 5 nm, (**b**) 10 nm and (**c**) 20 nm. (**d**) Relationship between the intensity of the R6G characteristic peak of 1508 cm^−1^ and the annealing temperature of Au films with various film thicknesses.

**Figure 6 molecules-28-01472-f006:**
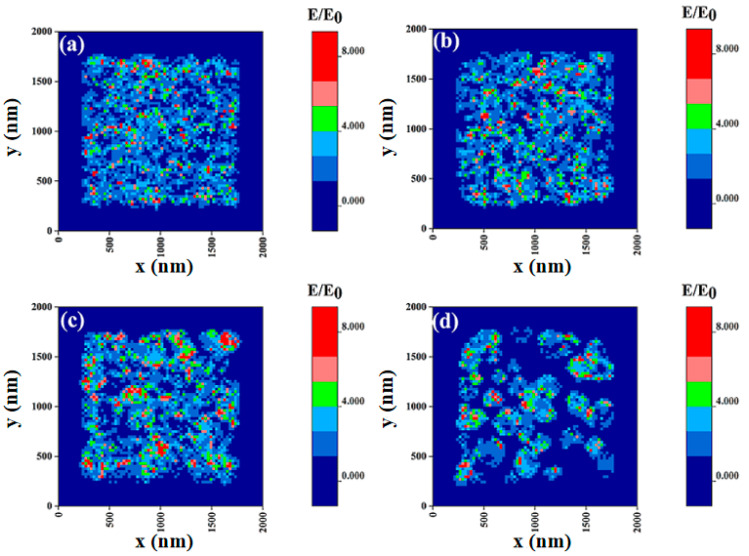
Electromagnetic field distribution of 10 nm Au films (**a**) without annealing and annealed at (**b**) 150 °C, (**c**) 250 °C and (**d**) 350 °C.

**Figure 7 molecules-28-01472-f007:**
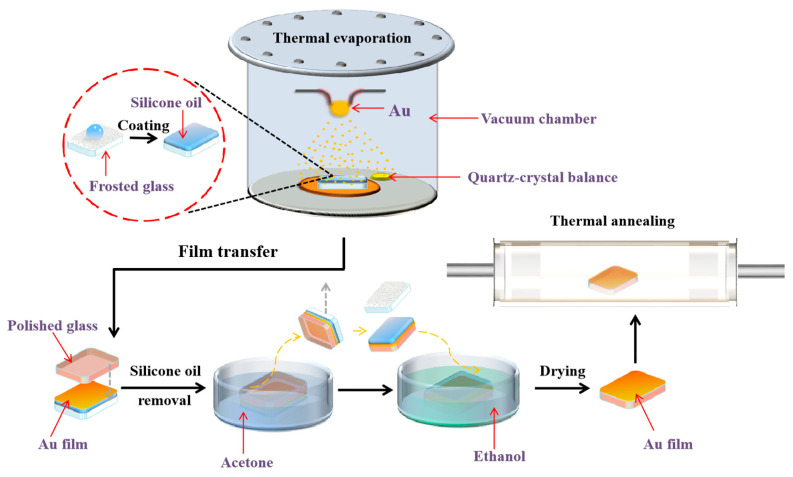
Schematic diagram of depositing Au atoms on silicone oil surfaces by thermal evaporation.

## Data Availability

Data available on request from the authors.

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
