# Peer review of "Thermal Annealing Effect on Surface-Enhanced Raman Scattering of Gold Films Deposited on Liquid Substrates"

_molecules, 2023, doi:10.3390/molecules28031472_

Round 1
Reviewer 1 Report
In this work, the authors prepared Au films with different thicknesses on liquid surfaces, which finally serve as the SERS substrates after the thermal treatment. The treatment parameters and morphologies were systematically studied to optimize the SERS performance. Along with the FDTD simulation results, the work can provide hints to design the SERS substrate in a facile and economic approach. I would recommend the acceptance after addressing the following concerns.
- For the liquid substrate, is there any substitute? If the viscosity has changed, how about the variation of the Au film morphology?
- Because both the Au diameter and the inter-gaps are important for SERS output, is it possible to control these two parameters separately?
- In Figure 4d-iv, is there any principle to describe such a “linear” relationship between the temperature and the diameter?
- As shown in Figure 6d, if the film thickness was further increased to be larger than 20 nm, will the sensing performance be better?
- Suggest to enhance the resolution of the SEM images.
Author Response
Response to Reviewer #1 Comments
Comment 1: For the liquid substrate, is there any substitute? If the viscosity has changed, how about the variation of the Au film morphology?
Answer:
Thanks for this reasonable question.
Yes, different kinds of liquid such as melted glass and ionic liquid have been used as liquid substrates in previous studies. In general, all of these liquid surfaces can serve as substrates with isotropic and free-sustained properties, and the metal atoms deposited on the liquid substrate are more likely to diffuse to form a rough film containing ramified aggregates, as depicted in the two-stage growth model. For the second question, the influence of viscosity variation on the morphology of metallic film has been previously reported[1]. The viscosity of liquid affects the diffusion of Au atoms on the surface. According to the formula of Brownian motion, D∝F-1 (D: diffusion coefficient; F: friction coefficient), the friction coefficient F increases with the increase of viscosity, which will affect the diffusion of Au atoms on the liquid surface. The morphology of ramified Au films would also be influenced, and the surface coverage of Au film with the same nominal thickness will decrease. Relative discussions have been provided in the revised manuscript (Please see p.1, last paragraph, p.2, second paragraph and p.4, first paragraph).
Comment 2:
Because both the Au diameter and the inter-gaps are important for SERS output, is it possible to control these two parameters separately?
Answer:
Many thanks for this helpful comment.
In our experiment, we investigated the experimental conditions mainly by controlling annealing temperature and film thickness to adjust the morphology of Au films, which will result in the change in both diameter and gap width of Au nanoparticles at the same time. Therefore, it is difficult to control the size and gap width of Au nanoparticles separately through this method. In future work, we will make attempt to enhance the control of experimental parameters during the growth process.
Comment 3:
In Figure 4d-iv, is there any principle to describe such a “linear” relationship between the temperature and the diameter?
Answer:
Thanks very much for this reasonable suggestion.
Generally, solid Au has a high melting point (~1064℃), and decreased significantly with the size of Au nanoparticles, especially for those with diameter smaller than 5nm. In our prepared Au films, the branched aggregates consisted of Au nanoparticles with a wide range of diameter. During the annealing process, Au nanoparticle with small diameter melted at low temperatures and merged with the adjacent large nanoparticles, the diameter of aggregates increased accordingly. On the other hand, the increasing size of Au nanoparticles further increased the melting point, resulting in an equilibrium between particle size and annealing temperature. Therefore, with the increase of annealing temperature, the diameter of Au nanoparticles increased continuously, showing a linear relationship as shown in Figure 4d-iv. We have revised the results and discussion section to explain the mechanism of the linear relationship between the annealing temperature and the diameter (Please see p.5, first paragraph).
Comment 4:
As shown in Figure 6d, if the film thickness was further increased to be larger than 20 nm, will the sensing performance be better?
Answer:
Thanks for this important comment.
It is possible to happen. In general, because of the characteristics of the liquid substrate, there are a large number of gaps within nanoparticles in Au films deposited on liquid surfaces, which can behave as “hot spots” in SERS measurement. According to the result shown in Figure 6d, since the surface coverage of Au film increases with film thickness, the branched Au aggregates will gradually connect with each other to form a continuous film. Therefore, it is speculated that if the film thickness continues to increase, the Au atoms will be further deposited on the continuous film which cannot be regarded as a free-sustained and isotropic substrate, which is beyond the range of discussion in our manuscript.
Comment 5:
Suggest to enhance the resolution of the SEM images.
Answer:
Thanks for this valuable suggestion.
We have revised the SEM images as required (Please see Figure 2, Figure 3, Figure 4 and Figure 5 in the revised manuscript).
Reviewer 2 Report
Thermal Annealing Effect on Surface-enhanced Raman Scattering of Gold Films Deposited on Liquid Substrates
Several inquiries for authors to clarify or revise:
1. Please include high-resolution figures. These figures are barely observable for details.
2. For the enhancement factor (EF), how to calculate the ?0 and ?????? From experimental measures or indirect estimation? Authors may need to put more details on these two quantities.
3. The surface tension of silicone oil plays an important role in the dispersion of Au thin films, are there any measurements on the surface tension? If there are such measurements then the correlation between the surface and morphology of Au films can be roughly estimated. But remember that the surface tension is sensitive to temperature change.
4. How can the 600C annealing be carried out? Isn’t there any evaporation of silicone oil? I think the thermodynamic change of the oil can dramatically influence the configuration of Au films. Also, another concern is the potential chemical reactions that could occur between the thin films and oil, can experimentally prove there is a stable co-existence of Au and silicone oil without any induced phase change at 600C?
5. The Raman spectra are indistinguishable, I am curious about the peaks of the main shift of Au or silicone oil. This part needs to wait until clear figures were provided.
6. Authors may need to identify different solid phases of Au at elevated temperatures and then compare them to the Raman spectra with some clear pictures of the annealing effects.
Author Response
Response to Reviewer #2 Comments
Comment 1:
Please include high-resolution figures. These figures are barely observable for details.
Answer:
Thanks very much for this important suggestion.
We have revised these figures as required (Please see Figure 2, Figure 3, Figure 4, Figure 5, Figure 6 and Figure 7 in the revised manuscript).
Comment 2:
For the enhancement factor (EF), how to calculate the N0 and NSERS? From experimental measures or indirect estimation? Authors may need to put more details on these two quantities.
Answer:
Thanks for this reasonable question.
and are the number of R6G molecules on the SERS substrate and a silicon wafer in the laser spot, respectively. Since it is difficult to determine the number of molecules enhanced in SERS measurement, we refer to previous methods to assume that probe molecules were uniformly distributed on the substrate, and estimate the number of probe molecules N that affect the signal by the following formula[2]:
where NA is Avogadro constant; C represents the molar concentration of R6G and V represents the volume of R6G solution; Abulk indicates the area of R6G distribution and Alaser is the area of laser spot. The preparation method of R6G samples adsorbed on the substrate and measured in the laser spot was identical, and the measurement conditions also were same. Therefore, (where is the concentrations of the R6G on the silicon wafer, indicates the concentrations of substrates). The EF was finally calculated as the ratio of relationship between concentration C and SERS intensity I, which is about 106.
The detailed calculation of enhancement factor has been provided in the revised manuscript as required (Please see p.7, last paragraph).
Comment 3:
The surface tension of silicone oil plays an important role in the dispersion of Au thin films, are there any measurements on the surface tension? If there are such measurements then the correlation between the surface and morphology of Au films can be roughly estimated. But remember that the surface tension is sensitive to temperature change.
Answer:
Thanks for this important question.
The surface tension of silicone oil substrate used in this experiment is about 36.5×10-2 J/m2 at room temperature, and it is sensitive to temperature change. The increase of temperature will reduce the surface tension, resulting in the change of the thickness of the silicone oil layer. It may result in uneven distribution of the liquid. The variation of the thickness of the silicone oil layer may affect the diffusion of Au clusters on the surface of the silicone oil, and thus affect the morphology of the Au film. Therefore, we tried to coat the silicone oil as flat as possible in the experiment. Then, in the deposition process, the sample was placed 25cm away from the evaporation source. In order to avoid damaging the Au film morphology in the annealing process and facilitate follow-up measurement, we first transferred the Au film to solid substrate and then carried out subsequent operations.
A discussion of the effect of silicone oil surface tension on the morphology of Au film was supplemented in the revised manuscript (Please see p.4, first paragraph).
Comment 4:
How can the 600°C annealing be carried out? Isn’t there any evaporation of silicone oil? I think the thermodynamic change of the oil can dramatically influence the configuration of Au films. Also, another concern is the potential chemical reactions that could occur between the thin films and oil, can experimentally prove there is a stable co-existence of Au and silicone oil without any induced phase change at 600°C?
Answer:
Thanks for this important question.
In order to avoid the influence of silicone oil during annealing, Au film has been transferred to a silicon wafer before annealing, and the residual silicone oil was removed by acetone and ethanol. Therefore, silicone oil will not influence the configuration of Au films during annealing.
Comment 5:
The Raman spectra are indistinguishable, I am curious about the peaks of the main shift of Au or silicone oil. This part needs to wait until clear figures were provided.
Answer:
Thanks very much for this helpful suggestion.
In this experiment, we used R6G as the probe molecule and mainly focused on the Raman intensity changes of several characteristic peaks of R6G in the Raman spectra. In the manuscript, we listed all the characteristic peaks of R6G, and judged whether it was affected by the characteristic peaks of other impurities by whether the peaks changed significantly with the experimental conditions. Since Au films compose of atoms, there is generally no Raman characteristic peak. Then, considering that the silicone oil may has not been thoroughly removed, the peaks of the main shift of 1120cm-1 and 1180cm-1 (the in-plane bending vibration modes of C-H) existing in silicone oil may affect the Raman detection. Therefore, we mainly compared and analyzed the signal enhancement intensity at 1508cm-1 (the C-C stretching vibration modes of aromatic ring). We have supplemented relevant discussion in the revised manuscript (Please see p.7, first paragraph).
Comment 6:
Authors may need to identify different solid phases of Au at elevated temperatures and then compare them to the Raman spectra with some clear pictures of the annealing effects.
Answer:
Thanks for this reasonable question.
As the morphology of Au films shown in the SEM images (Figure 4 and Figure 5), the ramified aggregates in Au film gradually disconnected and fractured when annealed at high temperature, then these irregular clusters gradually formed spherical Au nanoparticles. It has been found that the spheroidization process is easier to happen for thinner films with the increase of annealing temperature. According to the morphology of Au films in the SEM images under different annealing conditions, more spherical Au nanoparticles were produced under 150°C and 250°C annealing conditions, and part of small Au particles were merged into larger particles, which may be one of the reasons for the enhancement of Raman intensity. For the annealing temperature of 350℃, a significant increase in the gap width between the Au nanoparticles can be observed in the SEM images, thus we speculated that the gaps between Au nanoparticles exceed the distance of electromagnetic coupling which will cause the absence of partial nanoparticles couplings, leading to decreased SERS intensity.
More detailed discussion has been supplemented in the revised manuscript as required (Please see p.7, first paragraph).
Response to Reviewer #2 Comments
Comment 1:
Please include high-resolution figures. These figures are barely observable for details.
Answer:
Thanks very much for this important suggestion.
We have revised these figures as required (Please see Figure 2, Figure 3, Figure 4, Figure 5, Figure 6 and Figure 7 in the revised manuscript).
Comment 2:
For the enhancement factor (EF), how to calculate the N0 and NSERS? From experimental measures or indirect estimation? Authors may need to put more details on these two quantities.
Answer:
Thanks for this reasonable question.
and are the number of R6G molecules on the SERS substrate and a silicon wafer in the laser spot, respectively. Since it is difficult to determine the number of molecules enhanced in SERS measurement, we refer to previous methods to assume that probe molecules were uniformly distributed on the substrate, and estimate the number of probe molecules N that affect the signal by the following formula[2]:
where NA is Avogadro constant; C represents the molar concentration of R6G and V represents the volume of R6G solution; Abulk indicates the area of R6G distribution and Alaser is the area of laser spot. The preparation method of R6G samples adsorbed on the substrate and measured in the laser spot was identical, and the measurement conditions also were same. Therefore, (where is the concentrations of the R6G on the silicon wafer, indicates the concentrations of substrates). The EF was finally calculated as the ratio of relationship between concentration C and SERS intensity I, which is about 106.
The detailed calculation of enhancement factor has been provided in the revised manuscript as required (Please see p.7, last paragraph).
Comment 3:
The surface tension of silicone oil plays an important role in the dispersion of Au thin films, are there any measurements on the surface tension? If there are such measurements then the correlation between the surface and morphology of Au films can be roughly estimated. But remember that the surface tension is sensitive to temperature change.
Answer:
Thanks for this important question.
The surface tension of silicone oil substrate used in this experiment is about 36.5×10-2 J/m2 at room temperature, and it is sensitive to temperature change. The increase of temperature will reduce the surface tension, resulting in the change of the thickness of the silicone oil layer. It may result in uneven distribution of the liquid. The variation of the thickness of the silicone oil layer may affect the diffusion of Au clusters on the surface of the silicone oil, and thus affect the morphology of the Au film. Therefore, we tried to coat the silicone oil as flat as possible in the experiment. Then, in the deposition process, the sample was placed 25cm away from the evaporation source. In order to avoid damaging the Au film morphology in the annealing process and facilitate follow-up measurement, we first transferred the Au film to solid substrate and then carried out subsequent operations.
A discussion of the effect of silicone oil surface tension on the morphology of Au film was supplemented in the revised manuscript (Please see p.4, first paragraph).
Comment 4:
How can the 600°C annealing be carried out? Isn’t there any evaporation of silicone oil? I think the thermodynamic change of the oil can dramatically influence the configuration of Au films. Also, another concern is the potential chemical reactions that could occur between the thin films and oil, can experimentally prove there is a stable co-existence of Au and silicone oil without any induced phase change at 600°C?
Answer:
Thanks for this important question.
In order to avoid the influence of silicone oil during annealing, Au film has been transferred to a silicon wafer before annealing, and the residual silicone oil was removed by acetone and ethanol. Therefore, silicone oil will not influence the configuration of Au films during annealing.
Comment 5:
The Raman spectra are indistinguishable, I am curious about the peaks of the main shift of Au or silicone oil. This part needs to wait until clear figures were provided.
Answer:
Thanks very much for this helpful suggestion.
In this experiment, we used R6G as the probe molecule and mainly focused on the Raman intensity changes of several characteristic peaks of R6G in the Raman spectra. In the manuscript, we listed all the characteristic peaks of R6G, and judged whether it was affected by the characteristic peaks of other impurities by whether the peaks changed significantly with the experimental conditions. Since Au films compose of atoms, there is generally no Raman characteristic peak. Then, considering that the silicone oil may has not been thoroughly removed, the peaks of the main shift of 1120cm-1 and 1180cm-1 (the in-plane bending vibration modes of C-H) existing in silicone oil may affect the Raman detection. Therefore, we mainly compared and analyzed the signal enhancement intensity at 1508cm-1 (the C-C stretching vibration modes of aromatic ring). We have supplemented relevant discussion in the revised manuscript (Please see p.7, first paragraph).
Comment 6:
Authors may need to identify different solid phases of Au at elevated temperatures and then compare them to the Raman spectra with some clear pictures of the annealing effects.
Answer:
Thanks for this reasonable question.
As the morphology of Au films shown in the SEM images (Figure 4 and Figure 5), the ramified aggregates in Au film gradually disconnected and fractured when annealed at high temperature, then these irregular clusters gradually formed spherical Au nanoparticles. It has been found that the spheroidization process is easier to happen for thinner films with the increase of annealing temperature. According to the morphology of Au films in the SEM images under different annealing conditions, more spherical Au nanoparticles were produced under 150°C and 250°C annealing conditions, and part of small Au particles were merged into larger particles, which may be one of the reasons for the enhancement of Raman intensity. For the annealing temperature of 350℃, a significant increase in the gap width between the Au nanoparticles can be observed in the SEM images, thus we speculated that the gaps between Au nanoparticles exceed the distance of electromagnetic coupling which will cause the absence of partial nanoparticles couplings, leading to decreased SERS intensity.
More detailed discussion has been supplemented in the revised manuscript as required (Please see p.7, first paragraph).
Round 2
Reviewer 2 Report
I think the authors already answered most of the comments in the first round of review. Although there are still some scientific issues regarding the influence of the surface tension of silicone oil on the dispersion of Au films, I think at this point these issues should be addressed in future studies. For example, the thermal equilibrium of surface morphology of Au films interacting with silicone oil. How the finite thickness of films affects the dispersion of films by annealing. etc.
I recommend a possible publication but with checks on the writing.
Author Response
Many thanks for this helpful comment.
In this experiment, we transferred the Au film to the solid substrate in advance because the reduction of surface tension in the heating process would lead to the change of the thickness of the silicone oil layer and the evaporation of the silicone oil annealed at high temperature. At present, it is experimentally difficult to reach the thermal equilibrium between the Au film and silicone oil during annealing without damaging the morphology of the film. In the future, we will further study the effect of surface tension of silicone oil on the dispersion of the film, explore the most suitable temperature to optimize the characteristics of Au films on the surface of silicone oil, and investigate the relationship between its surface tension and the thickness of silicone oil.
We have checked the writing and added relevant discussion about this important issue in the revised manuscript (Please see p.9, first paragraph).